

**WFDEI-GEM-CaPA: A 38-year High-Resolution Meteorological Forcing Data Set for Land**
**Surface Modeling in North America**
**Zilefac Elvis Asong[1], Howard Simon Wheater[1], John Willard Pomeroy[1, 2], Alain Pietroniro[1, 2, 3],**
**Mohamed Ezzat Elshamy[1], Daniel Princz[1], and Alex Cannon[4]**
[1]*Global Institute for Water Security, University of Saskatchewan, 11 Innovation Blvd, Saskatoon, SK,*
*Canada S7N 3H5*
[2]*Centre for Hydrology, University of Saskatchewan, 121 Research Drive, Saskatoon, SK, Canada S7N 3C*
[3]*Environment and Climate Change Canada, 11 Innovation Blvd, Saskatoon, SK, Canada S7N 3H5*
[4]*Climate Research Division, Environment and Climate Change Canada, BC V8W 2Y2, Victoria, Canada*
**\*Corresponding author:**
Phone: +1 306 491 9565
Email: *elvis.asong@usask.ca*















**Abstract**:

Cold regions hydrology is very sensitive to the impacts of climate warming. Future warming is expected to increase the proportion of winter precipitation falling as rainfall. Snowpacks are expected to undergo less sublimation, form later and melt earlier and possibly more slowly, leading to earlier spring peak streamflow. More physically realistic and sophisticated hydrological models driven by reliable climate forcing can provide the capability to assess hydrologic responses to climate change. However, hydrological processes in cold regions involve complex phase changes and so are very sensitive to small biases in the driving meteorology, particularly in temperature and precipitation. Cold regions often have sparse surface observations, particularly at high elevations that generate a major amount of runoff. The effects of mountain topography and high latitudes are not well reflected in the observational record. The best available gridded data in Canada is from the high resolution forecasts of the Global Environmental Multiscale (GEM) atmospheric model and the Canadian Precipitation Analysis (CaPA) but this dataset has a short historical record. The EU WATCH ERA-Interim reanalysis (WFDEI) has a longer historical record, but has often been found to be biased relative to observations over Canada. The aim of this study, therefore, is to blend the strengths of both datasets (GEM-CaPA and WFDEI) to produce a less-biased long record product (WFDEI-GEM-CaPA). First, a multivariate generalization of the quantile mapping technique was implemented to bias-correct WFDEI against GEM-CaPA at 3h × 0.125° resolution during the 2005-2016 overlap period, followed by a hindcast of WFDEI-GEM-CaPA from 1979. The final product (WFDEI-GEM-CaPA, 1979-2016) is freely available at the Federated Research Data Repository (http://dx.doi.org/10.20383/101.0111).

**Subject Keywords:** cold regions processes, observations, bias correction, North America



**1    Introduction**

53        Accurate and reliable weather and climate information at watershed-to-basin scale is in

increasingly high demand by policy-makers, scientists, and other stakeholders for various purposes such
as water resources management (Barnett et al., 2005), infrastructure planning  (Brody et al., 2007), and
ecosystem modelling (IPCC, 2013). Particularly, the potential impacts of a warming climate on water
availability in snow-dominated high latitude regions continue to be a serious concern given that over the
past several decades, these regions have experienced some of the most rapid warming on earth (Demaria
et al., 2016; Diffenbaugh et al., 2012; Islam et al., 2017; Martin and Etchevers, 2005; Stocker et al., 2013).
The on-going science suggests that these warming trends are resulting in the intensification of the
hydrologic cycle, leading to substantial recent observed changes in the hydro-climatic regimes of major
river basins in North America (Coopersmith et al., 2014; DeBeer et al., 2016; Dumanski et al., 2015).
Changes in the timing and magnitude of river discharge (Dibike et al., 2016), shifts in extreme temperature
and precipitation regimes (Asong et al., 2016b; Vincent et al., 2015) and changes in snow, ice, and
permafrost regimes are anticipated (IPCC, 2013). Substantial evidence also indicates that the long-held
notion of stationarity of hydrological processes is becoming invalid in a changing climate.  As pointed out
by Milly et al. (2008), this loss of stationarity means that there will be an increase in the likelihood and
frequency of extreme weather and climate events, including floods, droughts, and heat and cold waves.

69        Water resources in most land areas north of 30° N are heavily dependent on natural water storage

provided by snowpacks and glaciers, with water accumulated in the solid phase during the cold season
and released in the liquid phase during warm events and the warm season. Particularly, the Rocky
Mountains, the hydrologic apex of North America with headwater streams flowing to the Arctic, Atlantic
and Pacific oceans, constitute an integral part of the global hydrologic cycle (Fang et al., 2013). Flows in
these high elevation headwaters depend heavily on meltwater from snowpacks and glaciers. However,
given that it is characterized by a highly varying cold region hydro-climate, studies indicate that it is in



these high elevation regions where climate variability and change is expected to be most pronounced in
terms of its impacts on water supply (Beniston, 2003; Kane et al., 1991; Prowse and Beltaos, 2002; Woo
and Pomeroy, 2011). More physically realistic and sophisticated hydrological models driven by reliable
climate forcing information can enhance our ability to assess short- and long-term regional hydrologic
responses to increasing variability and uncertainty in hydro-climatic conditions in a changing climate.
Nonetheless, hydrological processes in cold regions involve complex phase changes and so are very
sensitive to small biases in the driving meteorology, particularly in temperature and precipitation.

83        Cold regions often have sparse surface observations, particularly at the high elevations that

generate a major amount of runoff. The effects of mountain topography and high latitudes are currently
not well reflected in the observational record. Ground-based measurements (e.g. gauges) are limited
especially over the Rocky Mountains, and suffer from gross inaccuracies associated with cold climate
processes (Asong et al., 2017; Wang and Lin, 2015; Wong et al., 2017). The advent and use of weather
radar systems have addressed some of the short-comings of gauge coverage, at least where radar exists.
Unfortunately, in Canada, for example, the spatial coverage of weather radar is limited to the southern
(south of 55° N) part of the country (Fortin et al., 2015b). Recently, improved satellite products have
emerged such as the Global Precipitation Measurement (GPM) mission that provides meteorological
information at fine spatiotemporal resolutions and regular intervals. But, the GPM is still at its early stage
and only covers the region south of 60° N (Asong et al., 2017; Hou et al., 2014).

94        The capability of the current generation of Earth System Models (ESMs) to represent

meteorological variables is therefore of major interest for hydrological climate change impact studies in
cold regions watersheds. Despite substantial progress being made, raw outputs from regional and global
ESMs still differ largely from observational reference meteorology due partly to spatial scale mismatches
and systematic biases (Taylor et al., 2012). Therefore, ESM outputs are often downscaled and biases are
adjusted statistically before being used in hydrological simulations (Asong et al., 2016b; Chen et al., 2013;



Chen et al., 2018; Gudmundsson et al., 2012). Apart from uncertainty due to the many empirical statistical
techniques which have been developed to post-process ESM outputs (Maraun, 2016), the quality and
length of the reference observational data set for bias correction remains a major issue (Reiter et al., 2016;
Schoetter et al., 2012; Sippel et al., 2016). In Canada and other regions of North America, regional gridded
data sets such as the combined Global Environmental Multiscale (GEM) atmospheric model forecasts (Yeh
et al., 2002) and the Canadian Precipitation Analysis—CaPA (Mahfouf et al., 2007)  have been found to
perform comparably to ground observations, both statistically and hydrologically (Alavi et al., 2016;
Boluwade et al., 2018; Eum et al., 2014; Fortin et al., 2015a; Gbambie et al., 2017; Wong et al., 2017).
However, GEM-CaPA is too short to be used to directly correct ESM climate due to unsynchronized
internal variability—the recommended minimum record length for bias correction is 30 years  (Maraun,
2016; Maraun et al., 2017). Other gridded products such as the EU WATCH ERA-Interim reanalysis—WFDEI
(Weedon et al., 2014) and Princeton (Sheffield et al., 2006) have a longer historical record, but have been
found to be biased relative to observations over Canada (Wong et al., 2017) and the United States (Behnke
et al., 2016; Sapiano and Arkin, 2009). However, the WFDEI has been found to outperform other long-
record gridded products (Chadburn et al., 2015; Park et al., 2016; Wong et al., 2017).
The aim of this study, therefore, is to combine the strengths of both datasets (GEM-CaPA and
WFDEI) to produce a less-biased long record product (WFDEI-GEM-CaPA) using a multi-stage bias
correction framework. First, a multivariate generalization of the quantile mapping technique was
implemented to bias-correct WFDEI against GEM-CaPA at 3h × 0.125° resolution during the 2005-2016
period, followed by a hindcast of WFDEI-GEM-CaPA from 1979.
**2      Methodology**
**2.1     Data sources**
Hourly archived forecast data from the GEM model were acquired from Environment and Climate
Change   Canada   (http://collaboration.cmc.ec.gc.ca/cmc/cmoi/product_guide/submenus/rdps_e.html,





last access: 28 September 2018). The fields include downward incoming solar radiation, downward
incoming longwave radiation and pressure at the surface, as well as specific humidity, air temperature,
and wind speed at approximately 40 m above ground surface. The 40 m level was used because surface
variables (2 m temperature, 2 m specific humidity, and 10 m wind speed) are only available from 2010 in
the archive. The GEM data are approximately 24 km resolution from October 2001, approximately 15 km
from June 2004, and approximately 10 km resolution from November 2012, and are provided on a rotated
latitude/longitude grid in Environment and Climate Change Canada—ECCC 'standard file' format. The
archived data are of former operational forecasts, and contain model outputs from versions of GEM prior
to 2.0.0 through 5.0.0. A field for total precipitation (6-hourly) was acquired from the complementary
CaPA product (http://collaboration.cmc.ec.gc.ca/cmc/cmoi/product_guide/submenus/capa_e.html, last
access: 28 September 2018), which incorporates observed precipitation from meteorological weather
stations, and more recently from radar, into the precipitation field from GEM. The CaPA data are
approximately 10 km resolution from January 2002, also on a rotated latitude/longitude grid in ECCC
'standard file' format. The data contain reanalysis outputs from CaPA 2.4b8 from 2002-2012, and of
former operational analyses from versions of CaPA 2.3.0 through 4.0.0 from November 2012 onward. The
fields from GEM and CaPA were spatially interpolated and re-projected to a regular latitude/longitude
grid at 0.125° resolution. From GEM, they were interpolated using a bilinear algorithm, while CaPA was
interpolated using nearest neighbor (Schulzweida et al., 2004). Where necessary, GEM fields were
converted to SI units and CaPA was converted to a precipitation rate in SI units for better compatibility
with certain simulation models.

144         We also used the gridded WFDEI meteorological forcing data which has a global 0.5° spatial

resolution and 3-h time step covering the period 1979-2016 (http://www.eu-watch.org/data_availability,
last access: 25 July 2018). Weedon et al. (2014) used the ERA-Interim surface meteorology data as baseline
information to derive the WFDEI product. Firstly, ERA-Interim data were interpolated at half-degree



spatial resolution to match the land–sea mask defined by the Climatic Research Unit (CRU). Subsequently,
corrections for elevation and monthly bias of climate trends in the ERA-Interim fields were applied to the
interpolated data. The WFDEI data have two sets of precipitation data: the Global Precipitation
Climatology Centre product (GPCC) and CRU Time Series version 3.1 (CRU TS3.1). Thus, two variants of the
WFDEI product are available—WFDEI-GPCC and WFDEI-CRU. We used the WFDEI-CRU data set because it
goes up to 2016 while the WFDEI-GPCC had only been updated until 2013 at the time of our analysis.
**2.2      Data processing and bias correction workflow**

155           The workflow for the multi-stage bias correction is shown in Fig.1. Bias correction was done after

aggregating 1-h GEM-CaPA estimates to 3-h (the values at each time step represent the mean of the
previous 3-h period, to make it consistent with WFDEI) and interpolating both WFDEI and GEM-CaPA to
0.125° resolution. For bias correction, a multi-stage approach was implemented as follows. A multivariate
generalization of the quantile mapping technique (Cannon, 2018) which combines quantile delta mapping
(Cannon et al., 2015) and random orthogonal rotations to match the multivariate distributions of two data
sets was implemented to bias-correct WFDEI against GEM-CaPA at 3-h*0.125° resolution during the 2005-
2016 period. Models were fitted to data for each calendar month while accounting for inter-variable
dependence structure. Using the fitted models (2005-2016), a hindcast was made of WFDEI between
1979-2004. Finally, the corrected WFDEI data derived from the fitted (2005-2016) and hindcast (1979-
2004) periods were concatenated to obtain the bias-corrected WFDEI-GEM-CaPA product (1979-2016).


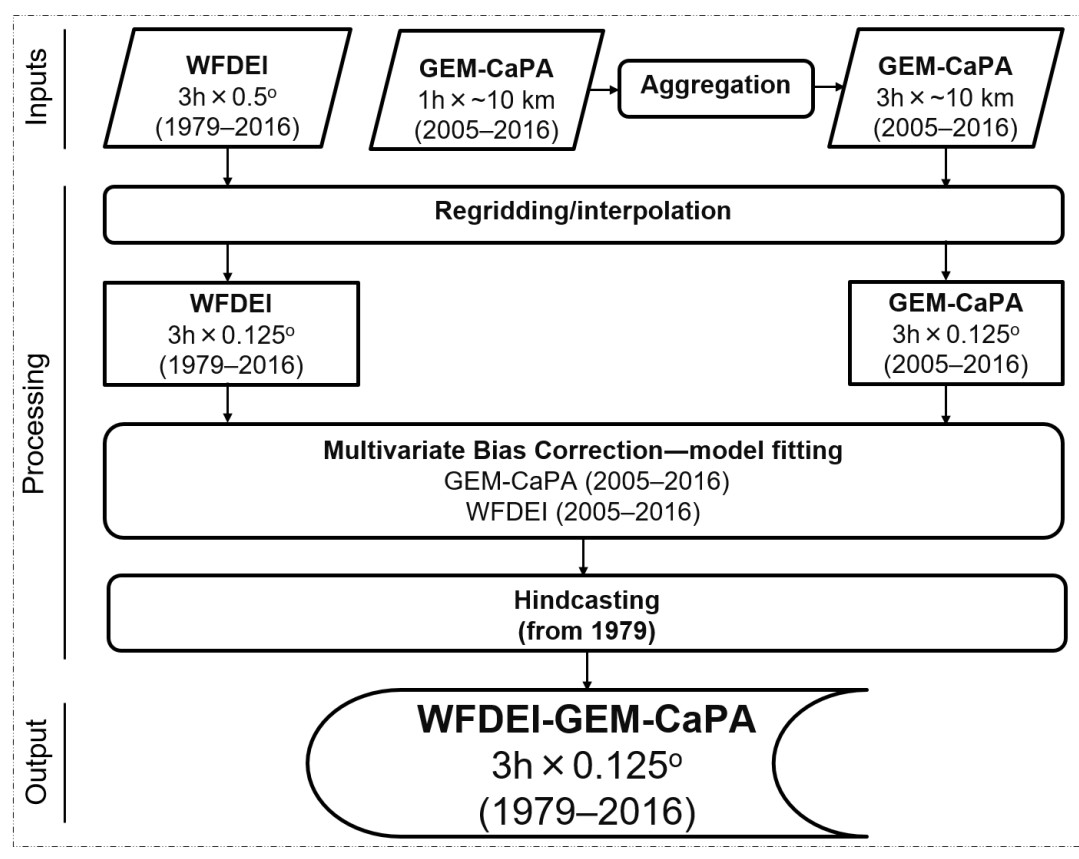

**Figure 1.** A schematic representation of inputs and bias correction procedure used to produce the WFDEI-GEM-CaPA meteorological forcing data set.

## 3    Results and discussion

Table 1 presents an overview of the seven variables processed in this study. Note that the GEM 40 m variables are used directly to correct WFDEI surface variables (2 m temperature, 2 m specific humidity, and 10 m wind speed). Therefore, the corrected WFDEI-GEM-CaPA data reflect 40 m variables. The spatial coverage of the WFDEI-GEM-CaPA data is depicted in Fig. 2. It spans the land region between longitude 50.0625° W to 149.9375° W and latitude 31.0625° N to 71.9375° N.





**Table 1:** List variables processed in this study with heights and units in each dataset.

| Variable | WFDEI Height | WFDEI Unit | GEM-CaPA Height | GEM-CaPA Unit | WFDEI-GEM-CaPA Height | WFDEI-GEM-CaPA Unit |
|---|---|---|---|---|---|---|
| Precipitation | Surface | kg m$^{-2}$ s$^{-1}$ | surface | kg m$^{-2}$ s$^{-1}$ | surface | kg m$^{-2}$ s$^{-1}$ |
| Air Temperature | 2 m | K | 40 m | K | 40 m | K |
| Specific Humidity | 2 m | kg kg$^{-1}$ | 40 m | kg kg$^{-1}$ | 40 m | kg kg$^{-1}$ |
| Wind Speed | 10 m | m s$^{-1}$ | 40 m | m s$^{-1}$ | 40 m | m s$^{-1}$ |
| Surface Pressure | Surface | Pa | Surface | Pa | Surface | Pa |
| Surface Downwelling Shortwave Radiation | Surface | W m$^{-2}$ | Surface | W m$^{-2}$ | Surface | W m$^{-2}$ |
| Surface Downwelling Longwave Radiation | Surface | W m$^{-2}$ | Surface | W m$^{-2}$ | Surface | W m$^{-2}$ |


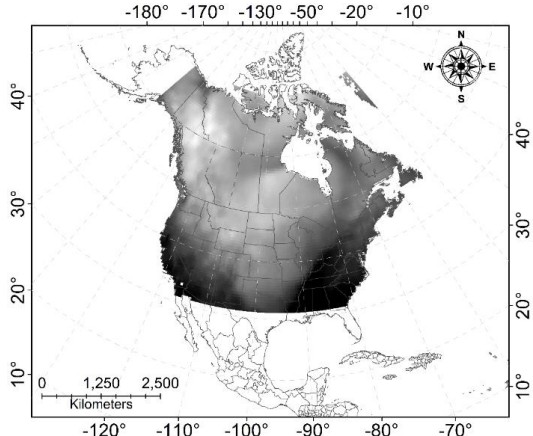


**Figure 2**: Spatial domain of the WFDEI-GEM-CaPA dataset spanning the region between longitude
50.0625° W to 149.9375° W and latitude 31.0625° N to 71.9375° N
The suitability of the bias correction algorithm to reproduce the observed annual cycle and inter-
annual variability of the variables was assessed for the fitting (2005-2016) and hindcast (1979-2004)
periods. Data extracted over the entire Mackenzie River basin is used to demonstrate the quality of the
bias correction exercise and uniqueness of the resulting output. Fig. 3 shows the annual cycle for GEM-
CaPA, WFDEI and WFDEI-GEM-CaPA during the fitting period. Overall, the monthly distributions show that
the bias was removed for all variables resulting in the very close distributions between GEM-CaPA and
WFDEI-GEM-CaPA. The bias was particularly large for wind speed, an important variable for both





mountainous and prairie hydrological processes, but was successfully removed. Fig. 4 shows the mean
annual time series of the seven variables over the 1979-2016 period. It is noticeable that the bias is
corrected while the inter-annual variability is well preserved between WFDEI and WFDEI-GEM-CAPA,
excerpt for shortwave radiation where the inter-annual variability is not fully preserved as shown by the
correlation between the WFDEI and WFDEI-GEM-CaPA annual series. However, this should not be a major
issue when impact models are forced with these data.

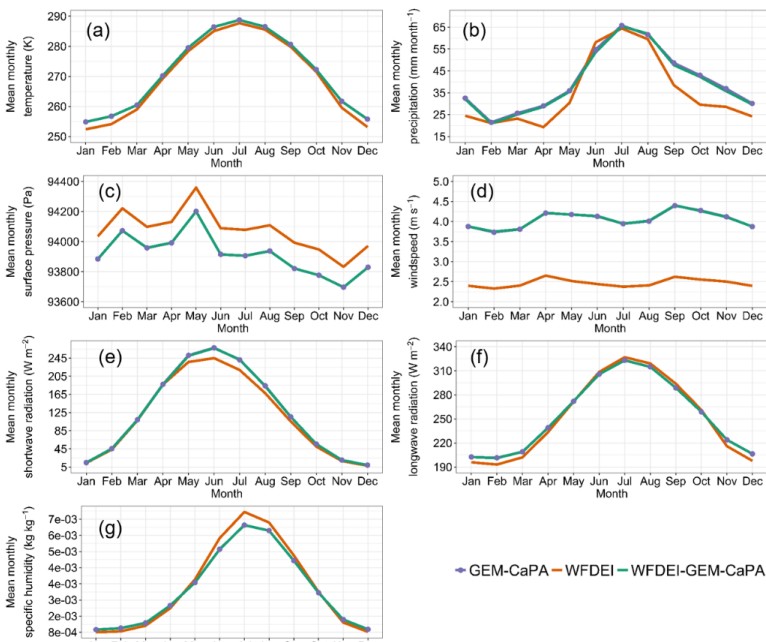

**Figure 3**: Annual cycle of GEM-CaPA (dark slate blue), WFDEI (orange) and bias corrected data—WFDEI-
GEM-CaPA (green) for air temperature (a), precipitation (b), surface pressure (c), wind speed (d),
shortwave radiation (e), longwave radiation (f), and specific humidity (g) during the fitting period (2005-

201    2016).



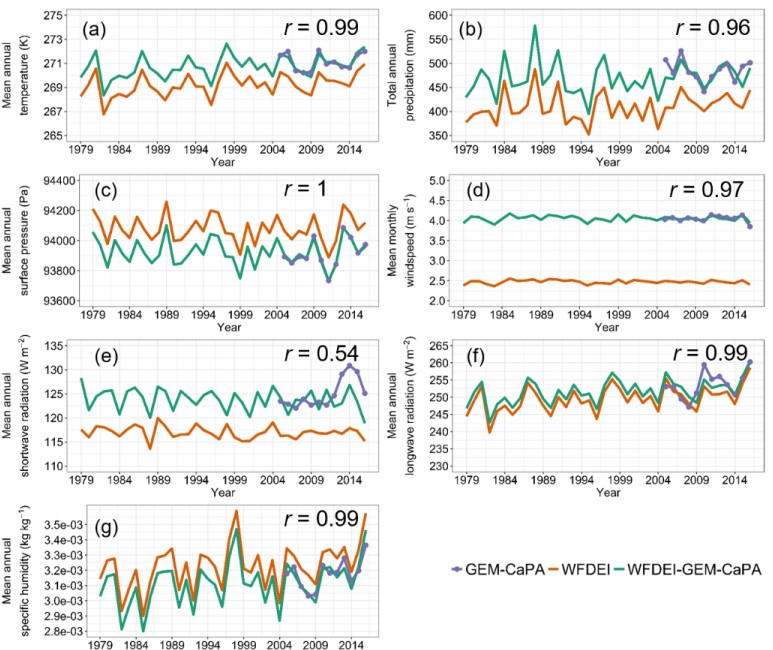


**Figure 4**: Time series of GEM-CaPA (dark slate blue), WFDEI (orange) and bias corrected data—WFDEI-

GEM-CaPA (green) for air temperature (a), precipitation (b), surface pressure (c), wind speed (d),

shortwave radiation (e), longwave radiation (f), and specific humidity (g) during the periods 2005-2016

(GEM-CaPA) and 1979-2016 (WFDEI and WFDEI-GEM-CaPA). The correlation ($r$) between the WFDEI and

WFDEI-GEM-CaPA annual series is indicated for each variable.

The foregoing analyses have shown that the bias in the WFDEI data was removed for both the

fitting and hindcast periods. However, some potential limitations remain—for example, WFDEI was

interpolated directly from 0.5° to 0.125° and bias-corrected against GEM-CaPA at 0.125°. The interpolation

does not add any event-scale spatial variability for a variable like precipitation which is very variable across

different scales. These issues have been reviewed extensively by (Cannon, 2018; Maraun, 2013; Maraun

et al., 2010; Storch, 1999).



**4      Conclusions**
Cold regions hydrology is very sensitive to the impacts of climate warming. More physically
realistic hydrological models driven by reliable climate forcing can provide the capability to assess
hydrologic responses to climate variability and change. However, cold regions often have sparse surface
observations, particularly at high elevations that generate a significant amount of runoff. By making
available this long-term dataset, we hope it can be used to better understand and represent the
seasonal/inter-annual variability of hydrological fluxes and the timing of runoff, and their long-term
trends. This unique data set will also prove valuable for bias correction of climate model projections to
assess potential impacts of future climate change on the hydrology and water resources of North America.
**5      Data availability**
The    latest    dataset    is    available    at    the    Federated    Research    Data    Repository
(http://dx.doi.org/10.20383/101.0111).
**Author contribution**
Z.E., H.W., J.P., A.P., and M.E. conceived of and designed the experiment. D.P. preprocessed the
GEM-CaPA data, A.C. developed the bias correction model code and guided the computing procedures
while Z.E. performed the simulations. M.E extracted the sample data used in generating Fig.3 and 4. Z.E.
prepared the manuscript with contributions from all co-authors.
**7      Competing interests**
The authors declare that they have no conflict of interest.
**8      Acknowledgements**
The financial support from the Canada Excellence Research Chair in Water Security, and Changing
Cold Regions Network is gratefully acknowledged. Thanks are due to the Meteorological Service of Canada
for providing access to the GEM-CaPA data used in this study. We also thank Dr. Graham Weedon for



making available the WFDEI data set. We also appreciate the efforts of Amber Peterson, Data Manager,
Global Institute for Water Security toward archiving the data at the Federated Research Data Repository.

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
