# Peer review of "WFDEI-GEM-CaPA: A 38-year High-Resolution Meteorological Forcing Data Set for Land Surface Modeling in North America"

_Earth System Science Data, 2018_

## Referee Comment (RC1) · Anonymous Referee #1 · 7 Nov 2018

This study developed a dataset by combining different datasets and bias correction method. Providing long-term dataset with high resolution is essential for land surface model; therefore, the dataset from this study is very important. The authors preliminarily validated the dataset in a watershed by comparing monthly and annual mean values. However, more detailed evaluation is necessary. First, similar validation for either monthly or annual means should be carried out for different regions with variable climatology. Second, the spatial pattern should be validated for those regions and periods with observations. Only with these validations, the reliability of the dataset can be concluded.

---

## Short Comment (SC1) · 16 Nov 2018

This study aimed to develop a long-term high-resolution dataset for North America by blending two existing datasets: WFDEI and GEM-CaPA. A multivariate generalization of the quantile mapping technique was use to bias-correct WFDEI against GEM-CaPA for seven meteorological variables. Mackenzie River basin was used as an example to illustrate the quality of the dataset. The dataset provided in this study and the quality assessment described in the manuscript have several fundamental issues and problems that needed to be vigorously addressed. The dataset did not deserve for publication in ESSD because of the following reasons: 1) Uniqueness of the dataset The

dataset provided in this study did not show any uniqueness at all. The authors merely used two existing datasets which are all publicly available and used an existing bias correction method to bias-correct the data. Uniqueness is one of the most important criteria in publishing in ESSD such that it is not possible to replicate the experiment or observation on a routine basis. In this regard, anyone in the world could actually repeat what the authors have done by downloading those datasets and applying the same bias correction method. In other words, the whole bias correction exercise in this study could be done on a routine basis. This is the fatal problem of the whole study.

2) Generation of meteorological variables It is very problematic to directly use GEM 40m variables to correct WFDEI surface variables [L173-175]. The two datasets estimate the same meteorological variables (e.g. temperature) at two different heights which essentially mean two different environmental conditions. As a result, two different statistical properties and distributions would be expected. By directly correcting one variable to another is like transforming an apple to an orange. The authors did not provide any justifications why this was done and did not discuss the implications of such bias-correction in these variables. In addition, the authors used two different methods to interpolate the data (i.e. bilinear for GEM whereas nearest neighbour for CaPA) [L140-141] and did not provide any justifications why such inconsistency existed when processing the data.

3) Usefulness of the dataset The usefulness of the data is significantly reduced because many land surface models (e.g. VIC) require near-surface temperature and wind speed as their meteorological inputs. However, those variables of the final product were corrected to 40m.This hinders potential users in using the final product because they have to find a way to re-process those variables back to reflect the near-surface conditions. Moreover, there are already other existing datasets covering North America, especially the United States, which are of long-term and high resolution. How useful will this dataset be as compared to the existing one when study areas are in the States, say Mississippi River basin for instance? In addition, the dataset did not truly covers the

whole North America and would not be useful for Alaska, Florida, and Texas because the spatial domain of the final product excludes part of those areas.

4) Validation of the data quality The authors had done a very poor job in assessing the quality of the data. The authors demonstrated the quality of the bias correction exercise by showing the annual cycle (Figure 3) and the mean annual time series of the datasets (Figure 4). This is not truly a quality assessment for the dataset. First of all, the final product is at 3-hr temporal resolution and checking its quality at annual time scale is a bit misleading because it is obvious that aggregating or averaging to a coarser resolution would provide a better performance regardless of the metrics used. The quality of the data should be assessed at its original temporal resolution in order to truly reflect the robustness of the data. Secondly, the final product covers North America but the authors only showed one basin as an example for quality assessment. There are at least several climatic zones across North America which consist of not only cold regions but also arid, semi-arid, and other regions. One basin in cold region is not sufficient to reveal the quality of the data. Several basins with different climatology should be used to assess the data quality. More importantly, the authors did not validate the final product with any observations. This is a very important and crucial step for any newly developed dataset that is generated by combining existing products. However, such assessment was not provided in the manuscript. This significantly reduces the reliability of the final product.

5) Scope of the study and dataset The dataset provided covers North America but the scope of the study discussed in the Introduction section was mainly focused on cold regions. The scope of the study did not match the dataset at all. The discussion in the Introduction section should be vigorously re-written by extending and including the importance of developing such dataset for other regions (e.g. desert) in addition to cold regions.

[Figure]

2018.

---

## Referee Comment (RC2) · Anonymous Referee #2 · 25 Nov 2018

Review ESSD-2018-128 Acronym soup reanalysis for Canada hydrology

The authors have provided a short clear outline of their combination of two Canada-focused forecast or forecast-amended-by-observation products (GEM and CaPA, respectively) with a global ECMWF reanalysis. However, the project and manuscript omit major steps necessary to make the data useful to readers and to qualify the description for ESSD. A valid data set for ESSD requires validation and quantitative uncertainty analysis. This manuscript presents neither.

First, from what we are given, evidently the demonstration outcome covers the "entire Mackenzie River basin" (line 186) but only that basin. First, does basin in this case equate to watershed? Or to a larger more general region defined by hydrography or ecology? Figures 3 and 4 apply only to this Mackenzie basin? Why only this basin? What does the outcome look like in eastern Canada? In the Canadian Rockies or BC? We need quantitative performance measures for multiple regions as well as for the entire area of coverage.

We do not get, but must have in order to develop confidence in the product, validation. We read about bias in the ECMWF product with respect to Canada (e.g. line 112) but for what parameters? Wind, precip, all parameters? Region-specific or Canada-wide? We never, however, find any attempt at validation of the combined product to a) show improvements over those previous but unspecified biases, or to b) validate this supposedly-improved product against regional and national observations. Authors chose Mackenzie basin because of number, quality and duration of observations? So, show us the bias-corrected product against real-world observations in that specific region. Also for agricultural regions of southern Canada? Canadian shield where Quebec hydro presumably has good long-term records? This manuscript joins others in mountain hydrology and western Canada cold regions hydrology special issues. Several of those data sets come from Canada, with sufficient temporal and spatial extent to serve as validation tests. Quite strange to find this product in those special issues with no apparent recognition of validation needs or possibilities, especially after the authors have made abundant mention of observational challenges in mountain regions. So, show us valid improvements using these quality-controlled well-described data sets mountainous regions.

We also get nothing on uncertainties. GEM comes with an error matrix. CaPA presumably reduces some of the precip errors of the forecast but adds its own errors from gauge observations and radar, particularly for snow and blowing snow. ERA-Interim likewise contains a host of well-known and well-documented uncertainties. Combining those three products, including changes in spatial resolution, time step and vertical extent (2m vs 40m) will have amplified the source uncertainties and imposed additional error terms. But the products presented in Table 1 and Figures 3 and 4 appear perfect, never a plus/minus or error bar among them. Especially for wind and precip, that simply can't be true? These authors would not themselves use this product in hydrological models without error statistics. How can they expect readers to plan any subsequent analysis or application sans uncertainty information? The manuscript requires an extensive quantitative assessment and discussion of uncertainties.